:ᦁ: PLOS | ONE

# Trueness of digital intraoral impression in reproducing multiple implant position

**Ryan Jin-Young Kim[1], Goran I. Benic[2], Ji-Man Park[3]\***

**1** Dental Research Institute, School of Dentistry, Seoul National University, Seoul, South Korea, **2** Clinic of Fixed and Removable Prosthodontics and Dental Material Science, Center for Dental Medicine, University of Zurich, Zurich, Switzerland, **3** Department of Prosthodontics, Yonsei University, College of Dentistry, Seoul, South Korea

\* jimarn@yuhs.ac

**Data Availability Statement:** All result data are available from the figshare database (accession number 10.6084/m9.figshare.9641996).

**Funding:** This work received a grant of the Korea Health Technology R&D Project from Korea Health

## Abstract

The aim of this study was to evaluate the trueness of 5 intraoral scanners (IOSs) for digital impression of simulated implant scan bodies in a partially edentulous model. A 3D printed partially edentulous mandible model made of Co-Cr with a total of 6 bilaterally positioned cylinders in the canine, second premolar, and second molar area served as the study model. Digital scans of the model were made with a reference scanner (steroSCAN neo) and 5 IOSs (CEREC Omnicam, CS3600, i500, iTero Element, and TRIOS 3) (n = 10). For each IOS's dataset, the XYZ coordinates of the cylinders were obtained from the reference point and the deviations from the reference scanner were calculated using a 3D reverse engineering program (Rapidform). The trueness values were analyzed by Kruskal-Wallis test and Mann-Whitney post hoc test. Direction and amount of deviation differed among cylinder position and among IOSs. Regardless of the IOS type, the cylinders positioned on the left second molar, nearest to the scanning start point, showed the smallest deviation. The deviation generally increased further away from scanning start point towards the right second molar. TRIOS 3 and i500 outperformed the other IOSs for partially edentulous digital impression. The accuracy of the CEREC Omnicam, CS3600, and iTero Element were similar on the left side, but they showed more deviations on the right side of the arch when compared to the other IOSs. The accuracy of IOS is still an area that needs to be improved.

## Introduction

With the aid of digital technology, traditional dental procedures are continuously being modified and optimized to become more convenient to both patients and clinicians. One of the most significant improvements in digital dentistry is the use of intraoral oral scanners (IOSs) for impression making. The use of IOSs allows to simplify the workflow for the fabrication of dental restorations by eliminating traditional polyvinyl siloxane impression and preparing stone dies in traditional method, thereby potentially reducing discomfort to patient, introduction of procedural errors and treatment time [1–3].

Since the advent of IOSs, its use has been accepted by many clinicians to adopt digital technology for acquisition of three-dimensional (3D) images of the dento-gingival tissues. For

Industry Development Institute (KHIDI) (https://www.khidi.or.kr/eps), funded by the Ministry of Health & Welfare (HI18C0435) to JMP. The funder had no role in study design, data collection and analysis, decision to publish, or preparation of the manuscript.

**Competing interests:** The authors have declared that no competing interests exist.

implant placement, IOS enables virtual planning with data from cone-beam computed tomography and fabrication of surgical guides for precise implant positioning. Impression of scan bodies using IOS digitally allows transferring the 3D position of the implant. Although deviation is inevitable during impression making regardless of the impression technique, impression has to be clinically accurate enough to allow fabricating a well-fitting restoration [4–6]. Misfit of implant-supported reconstructions may not only require more time for clinical adjustment but may also generate stress at the interface between the bone and implant as well as between the implant and prosthetic superstructure. Such stress could potentially cause detrimental biological and technical complications [7,8].

With regard to the accuracy between digital and conventional impression for implant-supported prostheses, controversy continues to exist. Some studies found superior [8,9], some similar [6,10,11], and other inferior [12–17] performance of digital impressions compared to that of conventional impression technique. In these studies, the accuracy of conventional impression was compared to that of digital impressions made by one [6,8–10,12–16] or two [11,17] types of IOSs. The accuracy of digital impression in partial or complete edentulous model for implant rehabilitation, albeit no consensus, has been compared among IOSs [18–26]. However, there is a lack of up-to-date information as to how various IOSs perform in terms of accuracy in digital implant impression. In addition, recent development of new scanning devices and technology and software upgrade warrants further investigation.

The purpose of this study was to evaluate the spatial accuracy of 5 IOSs in reproducing 6 bilaterally positioned simulated scan bodies in a partially edentulous model. The null hypothesis of this study was that that the accuracy of the digital impressions is not different between the IOSs and implant positions.

## Materials and methods

### Study model

To replicate a clinical scenario requiring a digital impression of the jaw after placing multiple scan bodies, on a mandibular partially edentulous model (E50-500 L; J. Morita Europe GmbH, Dietzenbach, Germany), canines, second premolars, and second molars were trimmed down bilaterally, leaving 1/5 of the cervical portion of the clinical crowns. A digital impression of the model was made with an industrial precision scanner (stereoSCAN neo; AICON 3D Systems GmbH, Braunschweig, Germany). A reverse engineering software (Rapidform; INUS Technology, Seoul, Korea) was used to virtually add a cylinder with a diameter of 2 mm and height of 7 mm on top of each of the 6 trimmed teeth. Three reference spheres with a diameter of 3.5 mm were added around the left second molar to set the reference three-dimensional coordinate system for the subsequent deviation measurement (Fig 1) [27]. Two spheres were positioned in the lingual aspect; one on the mesial and the other on the distal side of the left second molar, respectively. Another sphere was located in the distobuccal aspect of the left second molar cylinder to ensure that the coordinates of all the cylinders have positive values.

The cylinders were positioned perpendicular to the model axial plane, except for two cylinders on the left and right second molars, which were inclined 30 degrees mesially and distally, respectively. A master model made of cobalt-chromium (Co-Cr) was fabricated by a 3D printer (Eosint M270; EOS GmbH, Krailling, Germany) utilizing the direct metal laser sintering technology.

### Scanning procedure

The previously described industrial precision scanner was used to scan the 3D printed Co-Cr master model to obtain the reference dataset. Digital impressions of the master model were

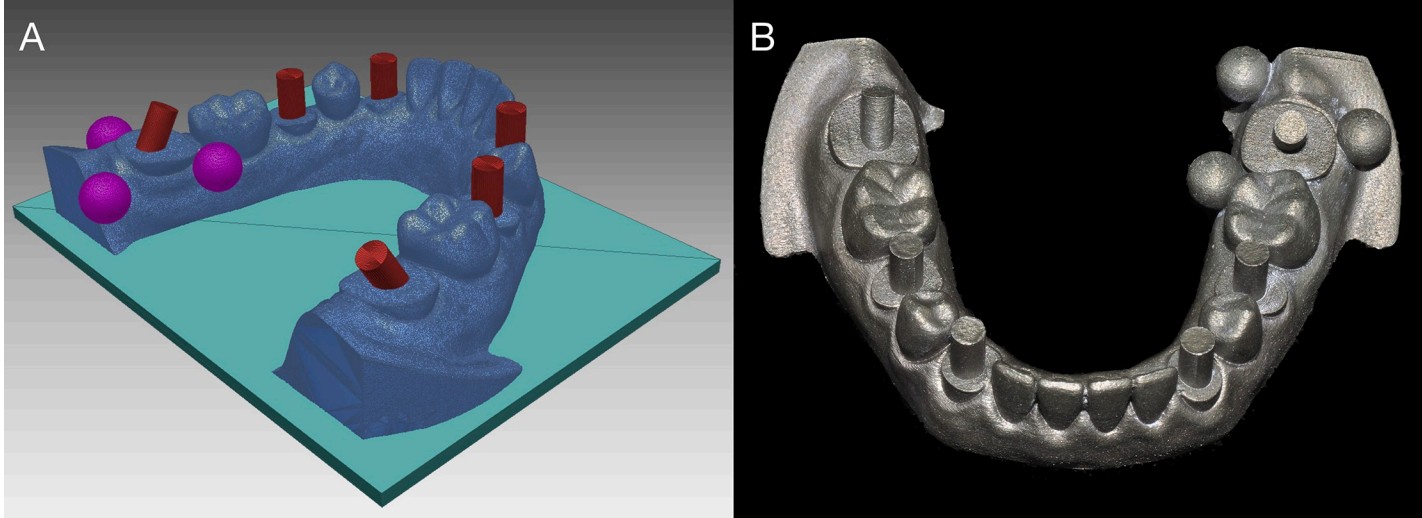

**Fig 1. Experimental model.** (A) Cylinders and reference spheres digitally formed using a reverse engineering software. (B) 3D printed Co-Cr master model.

performed using five IOSs (CEREC Omnicam (Dentsply Sirona, York, PA, USA), CS 3600 (Carestream Health, Rochester, NY, USA), i500 (Medit, Seoul, Korea)), iTero Element (Align Technology, San Jose, CA, USA), and TRIOS 3 (3Shape A/S, Copenhagen, Denmark)) (Table 1). For each scan, the spheres were scanned until no void was observed, and then the scanning procedures for the IOSs were performed along the occlusal surface starting from the left second molar to the right second molar, followed by the lingual and buccal side in the same experimental setting by an operator under ambient fluorescent lighting without the aid of additional lighting. No contrast powder was dusted prior to scanning. Additional scans were made to capture voided area of the cylinders that were critical for measurement. A total of 10 scans were performed by each IOS.

### Trueness evaluation of digital impression

The center of the reference sphere in the buccal aspect of the left second molar was set as the origin of the coordinate reference from which deviation of each cylinder was measured in the XYZ axes. The XY plane was formed by connecting the centers of the three spheres. The Y-axis was set as a line parallel to the line connecting the centers of the two spheres in the lingual aspect of the left second molar. The Y-axis denotes the anterior-posterior direction in the XY plane. The X-axis was set as a line perpendicular to the Y-axis, denoting the medial-lateral direction in the XY plane. The Z-axis denotes the coronal-cervical direction from the origin perpendicular to the XY plane.

**Table 1. Characteristics of intraoral scanners.**

| System | Manufacturer | Scanner technology | Light source | Acquisition method | Necessity of coating |
|---|---|---|---|---|---|
| CEREC Omnicam | Sirona Dental Systems | Active triangulation with strip light projection | Light | Video | None |
| CS3600 | Carestream Dental | Active triangulation (Stream projection) | Light | Video | None |
| i500 | MEDIT Corp. | Dual camera optical triangulation | Light | Video | None |
| iTero Element | Align Technologies | Parallel confocal microscopy | White LED light | Video | None |
| TRIOS 3 | 3shape | Confocal microscopy | Light | Video | None |

The reverse engineering software (Rapidform) was used to obtain the spatial information of the center of the top surface of cylinders in the form of XYZ coordinates from the reference origin for each scan. The coordinate distance between corresponding areas of the reference scan and each intraoral scan was then calculated to obtain the deviations, expressed either in positive or negative value, relative to the reference dataset. For each cylinder position, cumulative deviation in relation to the reference dataset was calculated by the root mean square of the overall XYZ values. The data were analyzed using SPSS Statistics for Windows, Version 23.0 (IBM Corp., Armonk. NY, USA). The Shapiro-Wilk test was carried out to verify the normality of each variable. The median trueness values of the IOSs were analyzed using the Kruskal-Wallis test, followed by Mann-Whitney U test and Bonferroni correction for pairwise comparisons (a = 0.05).

For visualization of the distribution of deviation of digital casts obtained by each IOS, an inspection software (Geomagic Verify v4.1.0.0; 3D Systems) was also used to superimpose the 3D digital casts acquired by the reference scanner and each IOS using a best fit algorithm.

## Results

The trueness values of the 5 IOSs at each cylinder position in XYZ axes and cumulative XYZ values are presented in Tables 2 and 3. Direction and magnitude of the deviation varied depending on the IOSs and the cylinder location (P<0.05). Regardless of the type of IOSs, there was a tendency for the median values and interquartile ranges to increase from the left second molar to the right second molar in the XYZ axes (P<0.05) (Fig 2, Table 2). The cumulative XYZ values were not significantly different between the IOSs (P = 0.101) (Table 3).

When the root mean square values of the overall XYZ values were pooled together, all the IOSs showed statistically significant gradual increase of the deviation from the left second molar to the right second molar (P<0.001) (Table 4). With respect to the overall trueness values, CS3600 showed the highest deviation, while i500 and TRIOS 3 outperformed the other IOSs. On the left side, TRIOS 3 was the only IOS that showed smaller deviation on the left second molar, but no significant difference in the trueness values were found at cylinders positioned on the left second premolar and left canine (P>0.05). The trueness values at cylinders positioned on the right side differed significantly among the 5 IOSs (P<0.05). The trueness values for CS3600 and CEREC Omnicam were similar to those obtained with iTero Element, i500, and TRIOS 3 on the left side from the second molar to the canine, while their deviation was greater on the right side towards the second molar position (Table 4).

Representative color-coded maps of digital casts obtained by each IOS are shown in Fig 3. The magnitude and direction of deviations on the color-coded map were not accurately matched with the XYZ deviations of the corresponding areas in Table 2.

## Discussion

Scan bodies have been used in the digital workflow of implant dentistry to supplant traditional impression procedure by digitally transferring the position of implant, saving cost and time for the clinicians and dental technicians, and reducing patient's discomfort during impression making [2,3,28]. New IOSs are being developed and have emerged on the market, while existing IOSs are also continuously being upgraded to a newer version of software to enhance their performance. The rising demand in digitalization by both dental team and patients is likely set the use of IOSs as the norm in routine daily practice after a satisfactory level of consensus on the application of IOSs for digital impression is clearly reached.

In this context, the present study was designed to clarify the performance of IOSs by evaluating the accuracy of 5 IOSs for acquisition of digital impressions of 6 simulated scan bodies

**Table 2. Trueness values (μm) of IOSs at each cylinder position in XYZ axes.**

| | | CEREC Omnicam | CS3600 | i500 | iTero Element | TRIOS 3 | Total | $\chi^2$ | df | P |
|---|---|---|---|---|---|---|---|---|---|---|
| X | 37 | 44.27 [-2.05, 54.64] | 29.93 [-4.02, 62.07] | -9.69 [-20.43, -6.31] | 60.40 [43.53, 83.24] | 28.31 [5.17, 50.38] | **60.38 B [22.45, 96.32]** | 16.274 | 5 | 0.006 |
| | 35 | -34.04 [-69.86, 17.08] | -103.48 [169.56, -40.46] | -87.04 [-106.17, -61.91] | -19.64 [-42.76, 11.68] | -38.25 [-95.91, 5.65] | **53.33 B [22.49, 100.51]** | | | |
| | 33 | -24.33 [-96.63, 72.20] | -158.14 [-282.30, -43.58] | -80.111 [-104.85, -53.12] | -78.61 [-106.65, -6.48] | -21.77 [-138.24, 61.29] | **88.01 AB [25.95, 194.09]** | | | |
| | 43 | 46.34 [-105.29, 165.76] | -174.68 [-429.17, -122.36] | -61.07 [-77.63, -22.82] | -129.30 [-176.21, -55.59] | 3.86 [-122.00, 82.15] | **64.56 AB [30.02, 129.27]** | | | |
| | 45 | 64.49 [-23.80, 115.87] | -142.42 [-295.17, -62.95] | -34.02 [-95.24, -4.09] | -81.62 [-103.70, -31.42] | -17.97 [-112.14, 63.33] | **80.88 AB [31.13, 158.88]** | | | |
| | 47 | 60.44 [-30.47, 312.78] | 29.02 [-153.51, 105.29] | 40.66 [-249.60, 137.04] | 24.93 [-80.93, 106.14] | -90.37 [-167.36, -23.73] | **105.69 A [59.20, 227.15]** | | | |
| | **Total** | **41.41 b [17.40, 105.36]** | **77.83 ab [40.17, 156.24]** | **88.58 a [51.60, 165.16]** | **64.82 ab [26.42, 124.65]** | **78.41 ab [21.86, 177.00]** | **69.51 [28.86, 142.89]** | | | |
| | $\chi^2$ | 10.147 | | | | | | | | |
| | df | 4 | | | | | | | | |
| | P | 0.038 | | | | | | | | |
| Y | 37 | 14.12 [-22.11, 44.87] | -12.56 [-28.48, 20.87] | -19.59 [-29.92, -11.64] | -56.55 [-69.82, -24.17] | 22.89 [20.34, 33.96] | **65.98 B [40.48, 105.78]** | 20.853 | 5 | 0.001 |
| | 35 | 11.02 [-28.33, 36.64] | -22.28 [-62.14, 6.25] | -22.32 [-43.29, -6.57] | -42.44 [-67.01, -26.88] | 1.17 [-8.60, 39.21] | **59.03 B [18.91, 183.38]** | | | |
| | 33 | -7.38 [-27.98, 46.98] | -42.71 [-83.77, 11.53] | -20.14 [-30.17, 0.86] | -59.03 [-80.04, -45.85] | -3.65 [-36.93, 26.94] | **94.72 AB [29.74, 254.86]** | | | |
| | 43 | 17.17 [-145.25, 193.83] | 115.47 [-5.02, 181.20] | -21.67 [-77.07, 81.03] | 71.05 [15.39, 127.09] | -98.72 [-157.67, 4.16] | **82.61 B [42.69, 174.36]** | | | |
| | 45 | 79.83 [-131.15, 310.77] | 244.59 [62.40, 289.60] | -11.02 [-88.75, 119.20] | 161.01 [58.17, 242.28] | -126.18 [-201.19, -8.47] | **133.91 AB [27.69, 223.38]** | | | |
| | 47 | 161.40 [-136.43, 407.79] | 279.26 [145.12, 370.96] | -39.42 [-109.46, 142.30] | 252.71 [63.21, 352.53] | -98.33 [-218.44, 42.29] | **175.79 A [88.72, 334.19]** | | | |
| | **Total** | **107.62 [51.12, 251.09]** | **81.13 [27.66, 168.70]** | **131.67 [35.49, 282.63]** | **42.45 [22.87, 207.94]** | **95.33 [56.28, 219.53]** | **91.76 [31.96, 218.34]** | | | |
| | $\chi^2$ | 6.537 | | | | | | | | |
| | df | 4 | | | | | | | | |
| | P | 0.162 | | | | | | | | |
| Z | 37 | 21.39 [-29.31, 31.10] | 20.80 [-13.44, 46.13] | 67.62 [57.45, 91.50] | 30.61 [-29.20, 41.26] | -6.15 [-17.02, -0.94] | **58.21 C [19.00, 110.76]** | 40.755 | 5 | <0.001 |
| | 35 | 26.68 [-12.20, 115.37] | 48.33 [-41.19, 99.41] | 86.05 [61.51, 107.44] | 87.21 [34.06, 144.88] | 38.44 [12.70, 97.62] | **30.39 C [17.52, 77.65]** | | | |
| | 33 | 40.03 [-47.37, 191.67] | 87.81 [-98.83, 131.79] | 119.57 [54.38, 200.77] | 205.35 [141.12, 261.26] | 117.37 [70.27, 239.92] | **90.45 A [54.66, 182.98]** | | | |
| | 43 | -87.47 [-394.93, 82.55] | -162.15 [-343.12, 249.95] | 89.63 [-39.50, 223.44] | 267.87 [198.11, 369.73] | -67.79 [-86.19, 86.03] | **94.97 AB [60.96, 149.63]** | | | |
| | 45 | -131.45 [-359.93, 31.13] | -173.70 [-484.47, 404.43] | 60.62 [5.57, 318.74] | 244.71 [206.13, 268.88] | -114.11 [-211.72, -48.55] | **66.69 B [26.22, 135.08]** | | | |
| | 47 | -257.54 [-437.34, -175.66] | -438.07 [-678.14, 444.85] | -33.51 [-222.06, 165.97] | 84.92 [42.15, 220.27] | -314.01 [-439.52, -208.12] | **125.22 A [71.31, 288.20]** | | | |
| | **Total** | **77.40 ab [30.17, 112.55]** | **67.30 b [24.52, 116.74]** | **64.45 b [25.32, 123.37]** | **91.59 ab [37.10, 174.70]** | **102.32 a [51.79, 249.28]** | **76.33 [31.05, 154.87]** | | | |
| | $\chi^2$ | 13.145 | | | | | | | | |
| | df | 4 | | | | | | | | |
| | P | 0.011 | | | | | | | | |

$\chi^2$, chi-square; df, degrees of freedom; P, P-value.

Interquartile ranges [1st quartile, 3rd quartile] are in parentheses.

Positive and negative values indicate deviation to the right and left in X-axis, forwards and backwards in Y-axis, upwards and downwards in Z-axis, respectively.

Absolute values were used for statistical analysis. Different uppercase letters within the same column indicate statistical difference between cylinder positions; different lowercase letters within the same row indicate statistical difference between IOSs (multiple comparison by Mann-Whitney U test with Bonferroni) (P<0.05).

**Table 3. Cumulative XYZ trueness values (μm) of IOSs.**

|  | CEREC Omnicam | CS3600 | i500 | iTero Element | TRIOS 3 | Total | $\chi^2$ | df | P |
|---|---|---|---|---|---|---|---|---|---|
| **X** | 27.10 [-61.05, 87.19] | -90.74 [-185.78, 17.15] | -50.44 [-87.87, -9.14] | -36.65 [-97.75, 56.91] | -13.98 [-103.14, 38.43] | **69.51 B [28.86, 142.89]** | 9.347 | 2 | 0.009 |
| **Y** | 16.79 [-30.44, 163.25] | 23.94 [-28.17, 182.71] | -21.58 [-46.76, 4.26] | -14.38 [-59.67, 144.22] | -10.35 [-107.81, 27.92] | **91.76 A [31.96, 218.34]** |  |  |  |
| **Z** | -7.69 [-203.85, 32.82] | 2.65 [-244.74, 97.09] | 74.85 [27.40, 149.50] | 150.21 [56.97, 257.13] | -15.03 [-117.20, 66.98] | **76.33 AB [31.05, 154.87]** |  |  |  |
| **Total** | **75.07 [25.97, 147.85]** | **72.20 [30.50, 158.58]** | **82.25 [38.20, 171.92]** | **68.52 [26.65, 155.05]** | **90.26 [43.22, 218.02]** | 78.45[30.91, 163.83] |  |  |  |
| **$\chi^2$** | 7.764 |  |  |  |  |  |  |  |  |
| **df** | 4 |  |  |  |  |  |  |  |  |
| **P** | 0.101 |  |  |  |  |  |  |  |  |

$\chi^2$, chi-square; df, degrees of freedom; P, P-value.

Interquartile ranges [1$^{st}$ quartile, 3$^{rd}$ quartile] are in parentheses.

Positive and negative values indicate deviation to the right and left in X-axis, forwards and backwards in Y-axis, upwards and downwards in Z-axis, respectively.

Absolute values were used for statistical analysis. Different uppercase letters within the same column indicate statistical difference between cylinder positions; different lowercase letters within the same row indicate statistical difference between IOSs (multiple comparison by Mann-Whitney U test with Bonferroni) (P<0.05).

that were bilaterally positioned in a partially edentulous model. To ensure the same testing condition, a commercially available assortment of artificial teeth that were screw-retained to a lower model was scanned, and the master model made of Co-Cr was fabricated by 3D additive manufacturing after modelling the cylinders to simulate screw-retained scan bodies. The dimensionally stable master model eliminated possible errors that could have occurred if external forces had been inadvertently applied to the screw-retained components during the experiment. The present study demonstrated that the accuracy of digital impressions varied significantly by IOSs and cylinder position. Therefore, the null hypothesis of this study that the IOS type and cylinder location would not affect the accuracy of digital impressions was rejected.

With regard to the cylinder position, deviation from true value was smallest at the cylinder located on the left second molar from which digital impression was sequentially made to the right second molar. Although some authors claimed that no significant differences in trueness were found between partially and completely edentulous implant models [22], arch length has been generally considered major culprit behind the development of deviation in a 3D virtual model due to the limited field of view of each capture using IOS. Captured multiple images are combined together by continuous stitching process at overlapping portion of the images, which is known to be the cause of deviation in a digitized model, processed by the proprietary software. This cumulative error accounts for the tendency for longer scanning span to generate greater chance of errors during the image combining process [4,19].

The overall accuracy was found to be best in the i500 and TRIOS 3 (Table 4). They also showed more consistent accuracy than the iTero Element, CEREC Omnicam and CS3600, which were, however, similar to the other IOSs on the left side from the second molar to the canine. The significantly greater range of trueness values were noted particularly in the CEREC Omnicam and CS3600 towards the opposite side of the origin of scanning. Within the limitations of the present study, the marked distortion on the right side suggests that the CEREC Omnicam and CS3600 may be well suited for unilateral partial-arch impression rather than for complete-arch scanning.

In a previous study that compared the accuracy of CEREC Omnicam, CS3600, TRIOS 3, and True Definition, CS3600 was found to be the best performing IOS [21]. The authors

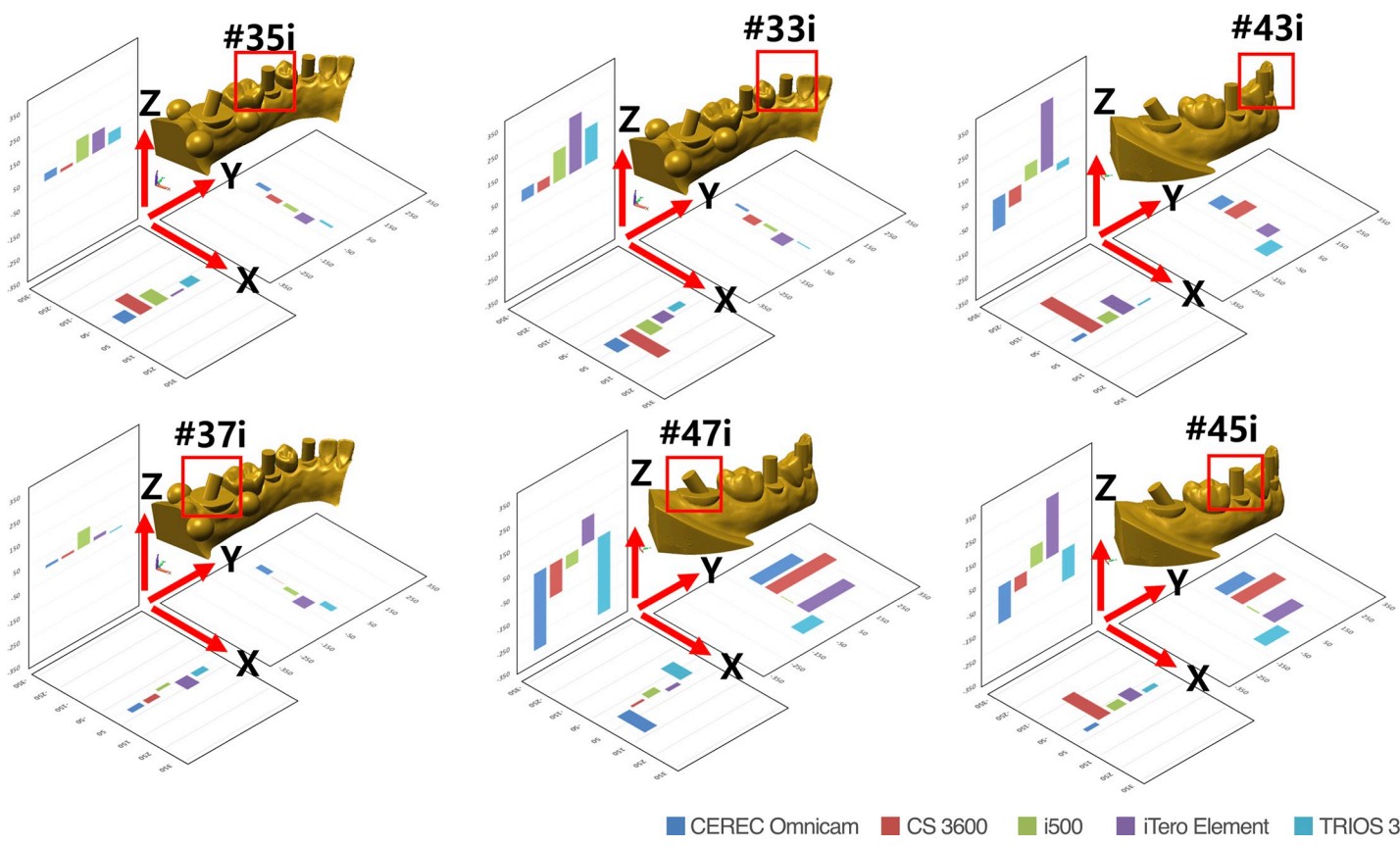

**Fig 2. Trueness values (μm) of IOSs at each cylinder position in XYZ axes.**

evaluated the entire trueness of each IOS for implant impression in a partially or a completely edentulous model using superimposing technique. The difference in the findings between this study and the present investigation might be explained by the different methodology employed for determining trueness. In this study, three reference spheres were required to form an XY plane, setting the reference coordinate system for three-dimensional deviation measurement. The reference spheres were positioned in close proximity to the left mandibular second molar area to enable them to be captured by IOSs at a time, thereby minimizing errors associated with image stitching process that could result by positioning them apart across the arch. Therefore, the present study measured the XYZ 3D displacement of the centroid of each cylinder. Thus, the XYZ deviations shown in Table 2 and Fig 2 were not accurately coincided with the corresponding areas on the color-coded map presented in Fig 3. The color-coded map is generated by superimposing datasets of the test group on to that of the reference scanner. Despite the fact that the color-coded map provides a general visual overview of scanning discrepancy by translating 3D deviation into 2D color-codes, superimposition by arbitrarily programmed best-fit may not be the most appropriate method in determining the trueness of IOSs at a specific location of interest. On the contrary to the previous studies that demonstrated only linear deviation [18–22], the XYZ coordinates used in the present study enabled precise acquisition of 3D spatial information of the individual cylinder by obtaining the differences of corresponding XYZ coordinates between the reference and test groups datasets. Direction and magnitude of the deviation in the XYZ axes varied depending on the IOSs and cylinder location. Insignificant differences in the cumulative XYZ total values among the IOSs

**Table 4. 3D root mean square deviation (μm) at each cylinder position.**

| | CEREC Omnicam | CS3600 | i500 | iTero Element | TRIOS 3 | $\chi^2$ | df | P | Total | $\chi^2$ | df | P |
|---|---|---|---|---|---|---|---|---|---|---|---|---|
| 37 | 75.42 Dab [58.78, 94.39] | 55.13 Dab [40.63–95.19] | 72.59 Da [58.23, 99.98] | 94.52 Ba [69.63, 116.85] | 51.40 Cb [41.50, 62.14] | 12.240 | 5 | 0.016 | 68.07 E [51.90, 94.45] | 168.887 | 5 | <0.001 |
| 35 | 123.98 CD [67.77, 157.67] | 172.91 CD [94.86–205.98] | 121.13 C [108.39, 154.38] | 100.21 B [59.63, 161.35] | 108.76 C [59.49, 123.96] | 5.640 | | 0.228 | 116.77 D [74.80, 158.06] | | | |
| 33 | 194.10 BCD [65.53, 273.40] | 209.79 BC [156.33–311.33] | 144.92 BC [122.79, 217.08] | 252.96 A [163.95, 340.71] | 171.73 B [136.27, 253.37] | 4.950 | | 0.293 | 187.16 C [145.59, 272.76] | | | |
| 43 | 289.09 ABCab [211.58, 443.48] | 403.42 ABa [194.03–886.19] | 204.33 ABCab [129.09, 288.62] | 314.61 Aa [267.09, 420.48] | 174.98 Bb [166.13, 207.32] | 16.794 | | 0.002 | 265.51 B [183.49, 401.81] | | | |
| 45 | 498.96 ABa [296.86, 1042.66] | 498.96 Aa [296.86–1042.66] | 232.14 ABab [146.77, 375.17] | 336.04 Aab [272.30, 388.25] | 212.64 ABb [156.77, 282.71] | 19.517 | | 0.001 | 322.01 AB [234.87, 497.31] | | | |
| 47 | 555.83 Aab [292.39, 647.73] | 670.89 Aa [472.81–1054.51] | 314.71 Ab [230.34, 518.94] | 343.99 Ab [168.54, 406.92] | 378.94 Ab [259.38, 514.09] | 13.416 | | 0.009 | 405.96 A [272.98, 585.61] | | | |
| $\chi^2$ | 32.280 | 40.788 | 36.294 | 38.481 | 36.867 | | | | | | | |
| df | | | 4 | | | | | | 195.33 [109.22, 357.99] | | | |
| P | <0.001 | <0.001 | <0.001 | <0.001 | <0.001 | | | | | | | |
| Total | 230.93 ab [94.39, 492.62] | 252.68 a [147.22, 532.77] | 150.34 b [109.63, 262.59] | 258.10 ab [117.43, 353.35] | 165.40 b [75.16, 245.09] | | | | | | | |
| $\chi^2$ | | | 16.885 | | | | | | | | | |
| df | | | 4 | | | | | | | | | |
| P | | | 0.002 | | | | | | | | | |

$\chi^2$, chi-square; df, degrees of freedom; P, P-value.

Interquartile ranges [1st quartile, 3rd quartile] are in parentheses.

Different uppercase letters within the same column indicate statistical difference between cylinder positions; different lowercase letters within the same row indicate statistical difference between IOSs (multiple comparison by Mann-Whitney U test with Bonferroni) (P<0.05).

(Table 3) was associated with the masking effect that yielded smaller cumulative deviation than the actual deviation due to the positive and negative values within the groups. The root mean square of the overall XYZ values were also calculated to directly compare the actual discrepancy of the digital impressions for each IOS. The findings of the present study were consistent with previous studies on the accuracy of digital implant impression that reported greater distortion with an increase in the scanning length [18–21,24–26].

IOS uses specific principle to acquire digital images of a real object. Although different data capture principles may be associated with the accuracy of IOS, based on the current literature, direct technique is deemed to provide more accurate impression as the number of implants increases [8,29]. But it cannot be asserted that the decrease in accuracy is directly attributable to the number of implants. Inaccurate digital impression in implant rehabilitation directly leads to mispositioning of virtual implant fixture which in turn may cause misfit of a fabricated prosthesis. From the biomechanical perspective, poorly fitting superstructures may be a detrimental factor to the longevity of restorations due to undue stress between the components [7,8].

For making impressions of a multiple angulated implant condition, digital impression could be a preferred approach given deformation of impression material during removal. As the angulation of implants increases, the impression material could be more distorted when removing it from the undercut areas. Nevertheless, the more implants that are being scanned, the longer the length of span that requires a greater number of images, theoretically resulting

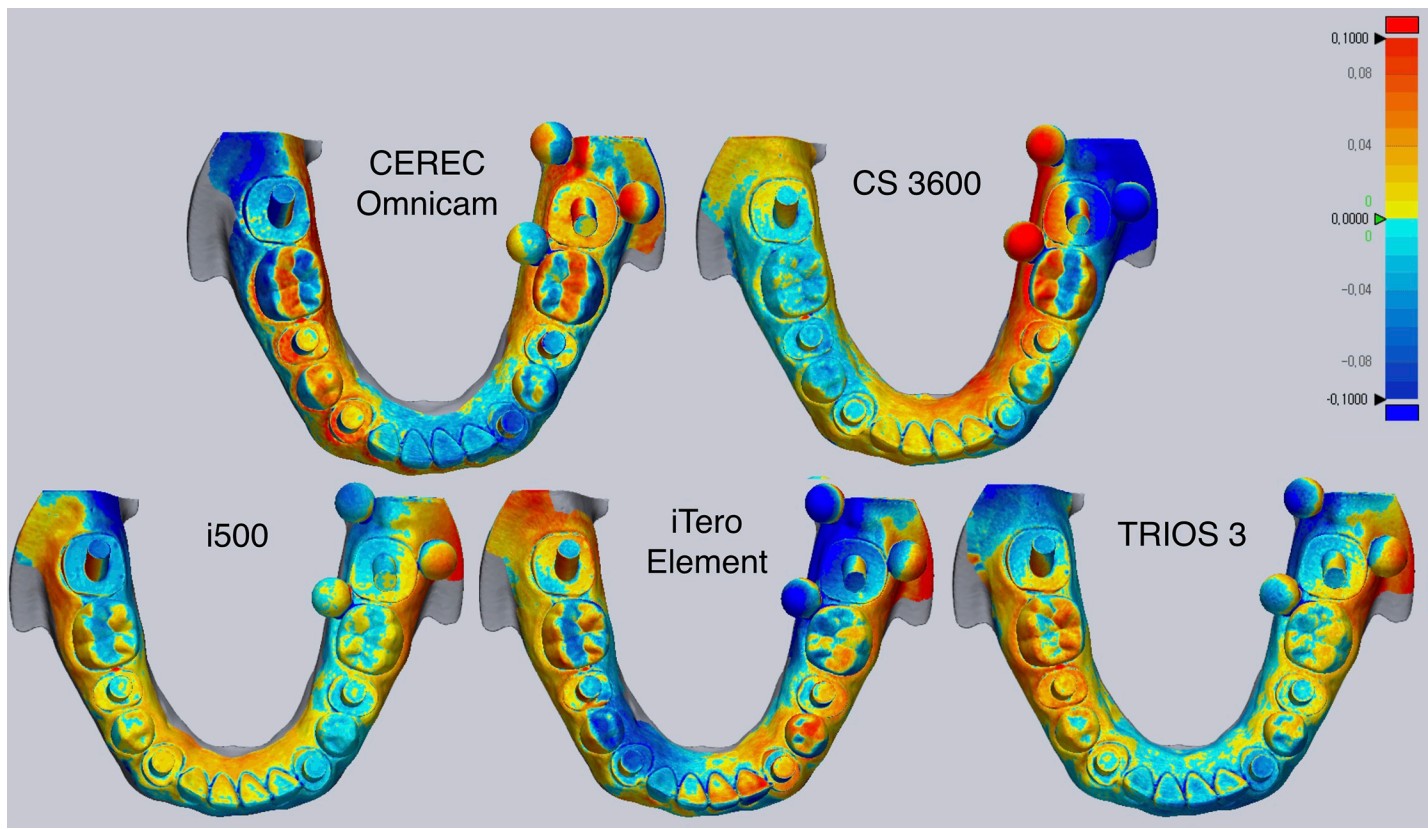

**Fig 3. Representative deviation of 3D digital casts.** Range of deviation is color-coded from −100 μm (blue) to +100 μm (red).

in a greater degree of cumulative errors. In our study model, accuracy of digital implant impression was evaluated in the dentate model, and this study confirmed that not all IOSs reproduced the same accuracy because of the differences in the data capture mode, principle, or software algorithms used in each IOS. This study also showed that some IOSs require further improvement to attain comparable accuracy. The size of the edentulous region should also be taken into consideration when investigating the accuracy of IOSs, since the lack of anatomic landmarks in smooth-surfaced soft tissue of edentulous region hinders proper superimposition of scans [30]. The inherent limitation of the present in vitro study includes that the experimental design does not represent a real clinical situation where the outcome is influenced by patient factors such as movement, soft and hard tissue interference, and moist condition caused by breathing and saliva secretion. The experimental model in this study had two tilted implants, one on each side of the rearmost area where there is a tendency for inexperienced dentists to install misaligned implants. Another limitation was that, despite a number of available scan bodies with various shapes and dimensions, only a single type of simulated scan bodies was used. Further studies should evaluate the influence of teeth or edentulous span, and different types and sizes of scan bodies to provide a better understanding of the accuracy of digital implant impression systems.

## Conclusions

Within the limitations of the present study, all the IOSs exhibited increasing deviation with an increasing distance from the start position of scanning. The direction and magnitude of

deviation differed among jaw regions and IOSs. All the IOSs were similar for unilateral arch scanning, while i500, and TRIOS 3 outperformed the other IOSs for partially edentulous scanning. The accuracy of IOS requires additional improvement.

## Acknowledgments

We thank J. Kim for technical assistance with sample preparation.

## Author Contributions

**Conceptualization:** Ji-Man Park.

**Data curation:** Ryan Jin-Young Kim, Ji-Man Park.

**Funding acquisition:** Ji-Man Park.

**Investigation:** Ryan Jin-Young Kim, Goran I. Benic, Ji-Man Park.

**Project administration:** Ji-Man Park.

**Validation:** Ryan Jin-Young Kim, Goran I. Benic, Ji-Man Park.

**Writing – original draft:** Ryan Jin-Young Kim, Ji-Man Park.

**Writing – review & editing:** Ryan Jin-Young Kim, Goran I. Benic, Ji-Man Park.

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
