## [Decision Letter · Decision Letter 0]

26 Sep 2019

PONE-D-19-23232

Trueness of digital intraoral impression in reproducing multiple implant position

PLOS ONE

Dear Pf. Park,

Thank you for submitting your manuscript to PLOS ONE. After careful consideration, we feel that it has merit but does not fully meet PLOS ONE’s publication criteria as it currently stands. Therefore, we invite you to submit a revised version of the manuscript that addresses the points raised during the review process.

Please revise the manuscript specifically on the terminology and statement. In particular, please clarify accuracy and precision which are most common confusing terms. Define the terms in your Methods and present your results and discussion accordingly. Also please address the Methods issue from Reviewer #2.

We would appreciate receiving your revised manuscript by Nov 10 2019 11:59PM. To enhance the reproducibility of your results, we recommend that if applicable you deposit your laboratory protocols in protocols.io, where a protocol can be assigned its own identifier (DOI) such that it can be cited independently in the future. For instructions see: http://journals.plos.org/plosone/s/submission-guidelines#loc-laboratory-protocols

We look forward to receiving your revised manuscript.

Kind regards,

Sompop Bencharit, DDS, MS, PhD, FACP

Academic Editor

PLOS ONE

Journal Requirements:

1. Please remove your figures from within your manuscript file, leaving only the individual TIFF/EPS image files, uploaded separately.  These will be automatically included in the reviewers’ PDF.

Additional Editor Comments (if provided):

There is a need to clarify several statements/terminologies per reviewers. Also please look into the study design (see Reviewer #2).

Reviewers' comments:

Reviewer's Responses to Questions

**Comments to the Author**

1. Is the manuscript technically sound, and do the data support the conclusions?

Reviewer #1: Yes

Reviewer #2: Yes

2. Has the statistical analysis been performed appropriately and rigorously? 

Reviewer #1: Yes

Reviewer #2: I Don't Know

3. Have the authors made all data underlying the findings in their manuscript fully available?

Reviewer #1: Yes

Reviewer #2: Yes

4. Is the manuscript presented in an intelligible fashion and written in standard English?

Reviewer #1: Yes

Reviewer #2: No

5. Review Comments to the Author

Reviewer #1: This benchtop study aims to investigate the accuracy of 5 intraoral digital scanning systems on digital impression. Multiple studies have used a similar design to evaluate digital impressions accuracy in-vitro, nevertheless, the study brings an interesting idea of multiple implants in a partially edentulous scenario. This study is somewhat new and bears scientific merit. However, there are some major issues that have to be resolved before acceptance for publication.

1. Please change the term “complete arch model” throughout the paper to “partially edentulous” because the term might be misunderstood as “fully edentulous”.

2. Although it is known that multiple images capturing and continuous stitching process at overlapping portion of the images can cause deviation in a digitized model, how do the authors evaluate the possible influence of the reference spheres located close to left second molar in the accuracy of obtained images?

3. Please define the cylinders size in this paper and discuss its impact on impression accuracy. Previous studies highlighted that different implant systems can lead to differences in optical and conventional impressions according to the characteristics of the scan bodies. Larger-size bodies can present more surface area for scanning and/or for impression materials.

4. Please discuss how the use of geometric device influence the outcomes seen in this paper. Geometric devices can offer landmarks that improve scanning accuracy, it especially important and was tested in large edentulous spaces.

5. I would suggest adding a paragraph for limitations of the study regards clinical translation since the in vitro design does not provide the best clinical scenario and implant positions (tilted distally), cylinders and spheres may also not represent a real clinical situation.

Other comments:

Abstract

• Page 2 Line 25

“simulated implant scan bodies in a complete-arch model”

- Seems to the reader it is a fully edentulous arches, specifying it is a partially edentulous model would be clearer to reader.

Material and Methods

• Page 4 Line 83

“add a cylinder with a dimeter of 2 mm” substitute with “diameter”

• Page 6 Table 1

Missing information for I500 intraoral scanner

Reviewer #2: Thank you for submitting your manuscript for review. Upon reviewing your manuscript, I found few points to bring to your attention.

First, there is a use of inappropriate dental terminology. For example, you should state “impression making” instead of “impression taking” throughout your article. There is also some grammatical error, for example, line 58 states: “Such stress could in turn potentially”, where the correct writing should “Such stress could, in turn, potentially” or “Such stress could potentially”. Also, you should change the sentence “eliminating the rubber based or alginate impressions” should be “eliminating traditional polyvinyl siloxane impressions and…” since rubber based material is almost obsolete in most dental practices nowadays. In addition, Table 1 has an error; you mention in your article that you used five scanners, the table has the details about four scanners only.

Second, in your discussion section, you mentioned that you evaluated precision. This wasn’t mentioned in the title or the aim of the manuscript. If you are going to report on precision it should be more clear and consistent throughout your manuscript.

In conclusion, the work in the manuscript would be a valuable contribution to the literature. My recommendation was to review your manuscript and make the recommended modifications.

6. PLOS authors have the option to publish the peer review history of their article (what does this mean?). If published, this will include your full peer review and any attached files.

Reviewer #1: No

Reviewer #2: No

---

## [Author Response · Author response to Decision Letter 0]

25 Oct 2019

Authors’ response to the reviewers’ comments

Re: Manuscript [PONE-D-19-23232] 

PLOS ONE 

Title: Trueness of digital intraoral impression in reproducing multiple implant position

We would like to extend our appreciation for taking the time and effort necessary to provide such insightful guidance. We have revised our paper and explained in response to the reviewers’ comments. We have exerted our best efforts to be completely responsive. We have included page and line numbers to help the reviewers keep track of the changes in the revised manuscript Word file with the ‘Track Changes’ function. We hope that these improve the paper such that the editor and the reviewers now deem it worthy of publication in PLOS ONE. 

REVIEWER #1

This benchtop study aims to investigate the accuracy of 5 intraoral digital scanning systems on digital impression. Multiple studies have used a similar design to evaluate digital impressions accuracy in-vitro, nevertheless, the study brings an interesting idea of multiple implants in a partially edentulous scenario. This study is somewhat new and bears scientific merit. However, there are some major issues that have to be resolved before acceptance for publication.

1. COMMENT 

Please change the term “complete arch model” throughout the paper to “partially edentulous” because the term might be misunderstood as “fully edentulous”.

RESPONSE: We are grateful for this comment as the term used to describe the experimental model could be misleading to the readers. We have changed from “complete-arch” to “partially edentulous model” throughout the manuscript.

(Page 2, Line 25, 37; Page 4, Line 78; Page 14, Line 209; Page 18, Line 313)

2. COMMENT 

Although it is known that multiple images capturing and continuous stitching process at overlapping portion of the images can cause deviation in a digitized model, how do the authors evaluate the possible influence of the reference spheres located close to left second molar in the accuracy of obtained images?

RESPONSE: We appreciate the reviewer’s observation that the manuscript lacks the description of the reference spheres. We have provided a brief description about the reference spheres in the Materials and Methods section. 

(Page 4, Lines 91-93) “Three reference spheres with a diameter of 3.5 mm were added around the left second molar to set the reference three-dimensional coordinate system for the subsequent deviation measurement”

We have also added more details in the Discussion section about the positioning and role of the reference spheres in this study. 

(Page 15, Lines 245-249) “Three reference spheres were required to form an XY plane, setting the reference coordinate system for three-dimensional deviation measurement. The reference spheres were positioned in close proximity to the left mandibular second molar area to enable them to be captured by IOSs at a time, thereby minimizing errors associated with image stitching process that could result by positioning them apart across the arch.” 

3. COMMENT 

Please define the cylinders size in this paper and discuss its impact on impression accuracy. Previous studies highlighted that different implant systems can lead to differences in optical and conventional impressions according to the characteristics of the scan bodies. Larger-size bodies can present more surface area for scanning and/or for impression materials.

RESPONSE: The 6 bilaterally positioned cylinders used in this study had the same dimensions of 2 mm diameter, 7 mm height. We have defined the cylinder size in the Materials and Method section. 

(Page 4, Line 89-91) “A reverse engineering software (Rapidform; INUS Technology, Seoul, Korea) was used to virtually add a cylinder with a dimeter of 2 mm and height of 7 mm on top of each of the 6 trimmed teeth.”

We agree with the reviewer that larger-size scan bodies offer a greater surface area for scanning and the impression outcome could differ depending on the type of implant systems and characteristic of the scan bodies. However, we used the same size and shape for the 6 simulated scan bodies in order to solely focus on the deviation of the digital impressions obtained by each IOS at various positions along the partially edentulous model. 

Nonetheless, as the reviewer pointed out, we have revised to clarify this limitation of the study in the Discussion section. 

(Page 17, Line 294 – Page 18, Line 303) “The inherent limitation of the present in vitro study includes that the experimental design does not represent a real clinical situation where the outcome is influenced by patient factors such as movement, soft and hard tissue interference, and moist condition caused by breathing and saliva secretion. The experimental model in this study had two tilted implants, one on each side of the rearmost area where there is a tendency for inexperienced dentists to install misaligned implants. Another limitation was that, despite a number of available scan bodies with various shapes and dimensions, only a single type of simulated scan bodies was used. Further studies should evaluate the influence of teeth or edentulous span, and different types and sizes of scan bodies to provide a better understanding of the accuracy of digital implant impression systems.”

4. COMMENT 

Please discuss how the use of geometric device influence the outcomes seen in this paper. Geometric devices can offer landmarks that improve scanning accuracy, it especially important and was tested in large edentulous spaces.

RESPONSE: We thank the reviewer for bringing up this point. We have clarified the role of the geometric device in this experimental study, as described above in the RESPONSE to COMMENT #2. 

With regard to the importance of landmarks, particularly in large edentulous spaces, we have provided the in the Discussion section. 

(Page 17, Line 291-294) “The size of the edentulous region should also be taken into consideration when investigating the accuracy of IOSs, since the lack of anatomic landmarks in smooth-surfaced soft tissue of edentulous region hinders proper superimposition of scans [30].”

5. COMMENT 

I would suggest adding a paragraph for limitations of the study regards clinical translation since the in vitro design does not provide the best clinical scenario and implant positions (tilted distally), cylinders and spheres may also not represent a real clinical situation.

RESPONSE: We have taken the reviewer’s suggestion and revised a paragraph in the Discussion section. 

(Page 17, Line 294 – Page 18, Line 303) “The inherent limitation of the present in vitro study includes that the experimental design does not represent a real clinical situation where the outcome is influenced by patient factors such as movement, soft and hard tissue interference, and moist condition caused by breathing and saliva secretion. The experimental model in this study had two tilted implants, one on each side of the rearmost area where there is a tendency for inexperienced dentists to install misaligned implants. Another limitation was that, despite a number of available scan bodies with various shapes and dimensions, only a single type of simulated scan bodies was used. Further studies should evaluate the influence of teeth or edentulous span, and different types and sizes of scan bodies to provide a better understanding of the accuracy of digital implant impression systems.”

6. OTHER COMMENTS

Abstract: Page 2 Line 25

“simulated implant scan bodies in a complete-arch model”

- Seems to the reader it is a fully edentulous arches, specifying it is a partially edentulous model would be clearer to reader.

RESPONSE: We have changed from “complete-arch” to “partially edentulous model” as mentioned in the RESPONSE to COMMENT #1. 

(Page 2, Line 25, 37; Page 4, Line 78; Page 14, Line 209; Page 18, Line 313)

Material and Methods: Page 4 Line 83

“add a cylinder with a dimeter of 2 mm” substitute with “diameter”

RESPONSE: We have corrected the typo. 

(Page 4, Line 90)

• Page 6 Table 1

Missing information for I500 intraoral scanner

RESPONSE: We apologize for the omission. We have added missing information of i500 IOS in Table 1. 

(Page 6, Line 125)

System / Manufacturer / Scanner technology / Light source / Acquisition method / Necessity of

coating

i500 / MEDIT Corp. / Dual camera optical triangulation / Light / Video / None

REVIEWER #2

Thank you for submitting your manuscript for review. Upon reviewing your manuscript, I found few points to bring to your attention.

1. COMMENT 

There is a use of inappropriate dental terminology. For example, you should state “impression making” instead of “impression taking” throughout your article. There is also some grammatical error, for example, line 58 states: “Such stress could in turn potentially”, where the correct writing should “Such stress could, in turn, potentially” or “Such stress could potentially”. Also, you should change the sentence “eliminating the rubber based or alginate impressions” should be “eliminating traditional polyvinyl siloxane impressions and…” since rubber based material is almost obsolete in most dental practices nowadays. In addition, Table 1 has an error; you mention in your article that you used five scanners, the table has the details about four scanners only.

RESPONSE: We sincerely appreciate the reviewer’s valid points. We have accordingly corrected the inappropriate dental terminologies/phrases and grammatical error.

1. “impression making” instead of “impression taking” 

(Page 3, Line 48, 57; Page 14, Line 202)

2. “Such stress could potentially” instead of “Such stress could in turn potentially” 

(Page 3, Line 61)

3. “eliminating traditional polyvinyl siloxane impression” instead of “eliminating the rubber based or alginate impression” 

(Page 3, Line 49)

Regarding Table 1, we have added missing information of i500 IOS. 

(Page 6, Line 125)

System / Manufacturer / Scanner technology / Light source / Acquisition method / Necessity of

coating

i500 / MEDIT Corp. / Dual camera optical triangulation / Light / Video / None

2. COMMENT 

In your discussion section, you mentioned that you evaluated precision. This wasn’t mentioned in the title or the aim of the manuscript. If you are going to report on precision it should be more clear and consistent throughout your manuscript.

RESPONSE: We appreciate this comment, which enabled us to detect unclear phrases. We have deleted the sentence “In terms of precision, which indicates the degree to which images acquired by repeated scanning are identical, the range of trueness values could be used to deduce the precision of each IOS.” to avoid possible confusion. 

3. COMMENT 

In conclusion, the work in the manuscript would be a valuable contribution to the literature. My recommendation was to review your manuscript and make the recommended modifications.

RESPONSE: The manuscript has been revised to reflect the reviewer’s suggested modifications. Thank you for the helpful comments and consideration of this revised manuscript for publication in PLOS ONE.

---

## [Editor Report · Decision Letter 1]

29 Oct 2019

Trueness of digital intraoral impression in reproducing multiple implant position

PONE-D-19-23232R1

Dear Dr. Park,

We are pleased to inform you that your manuscript has been judged scientifically suitable for publication and will be formally accepted for publication once it complies with all outstanding technical requirements.

With kind regards,

Sompop Bencharit, DDS, MS, PhD, FACP

Academic Editor

PLOS ONE

Additional Editor Comments (optional):

Thank you for the revision.
---

## [Editor Report · Acceptance letter]

1 Nov 2019

PONE-D-19-23232R1 

Trueness of digital intraoral impression in reproducing multiple implant position 

Dear Dr. Park:

I am pleased to inform you that your manuscript has been deemed suitable for publication in PLOS ONE. Congratulations! Your manuscript is now with our production department. 

With kind regards,

on behalf of

Dr. Sompop Bencharit 

Academic Editor

PLOS ONE